# Investigation of the Electronic Properties of Silicon Carbide Films with Varied Si/C Ratios Annealed at Different Temperatures

**Dan Shan [1,2,\*], Daoyuan Sun [2], Menglong Wang [2] and Yunqing Cao [2]**

1   School of Information Engineering, Carbon Based Low Dimensional Semiconductor Materials and Device Engineering Research Center of Jiangsu Province, Yangzhou Polytechnic Institute, Yangzhou 225127, China

2   Institute of Optoelectronic Technology, College of Physical Science and Technology, Yangzhou University, Yangzhou 225009, China; mz120210604@stu.yzu.edu.cn (D.S.); mz120210608@stu.yzu.edu.cn (M.W.); yqcao@yzu.edu.cn (Y.C.)

\*   Correspondence: shand@ypi.edu.cn; Tel.: +86-136-6526-3910

**Abstract:** Hydrogenated amorphous SiC (*a*-SiC:H) films with various Si/C ratios were prepared using the plasma-enhanced chemical vapor deposition (PECVD) technique. These films were then subjected to thermal annealing at different temperatures to induce crystallization. The electronic properties of the annealed SiC films were investigated through temperature-dependent Hall mobility measurements. It was found that the room-temperature Hall mobilities in the SiC films increased with both the annealing temperature and the Si/C ratio. This increase was attributed to the improved crystallization in the SiC films. Importantly, SiC films with different Si/C ratios annealed at different temperatures exhibited varying temperature dependence behaviors in their Hall mobilities. To understand this behavior, a detailed investigation of the transport processes in SiC films was carried out, with a particular emphasis on the grain boundary scattering mechanisms.

**Keywords:** SiC film; electronic property; temperature-dependent Hall mobility; grain boundary scattering mechanism

## 1. Introduction

In current research, increasing attention is being directed towards SiC films due to their applications in nanoelectronic and optoelectronic devices, including Si-based solar cells, Si-based light-emitting diodes, nonvolatile memory and biosensors [1–6]. It is generally accepted that SiC is beneficial for improving the performance of devices because it has a lower bandgap than $SiO_2$ or $SiN_x$ [7]. This lower bandgap contributes to improved carrier transport properties, leading to enhanced device performance. SiC is also a cost-effective semiconductor material with exceptional optical performance. According to Derst's research, SiC has high optical absorption in the photon energy region below the bandgap [8]. Rahul Pandey employed a strategy of designing hydrogenated amorphous SiC (*a*-SiC:H) to contact Si-based solar cells after passivation, taking into account both optical and electrical properties [9]. These approaches have proven to be effective ways to create efficient and reliable solar cells. Cao et al. systematically investigated the photovoltaic properties of Si nanocrystal (Si NC)/SiC multilayers. They prepared size-controllable solar cells based on Si NC/SiC multilayers and achieved a maximum power conversion efficiency (PCE) of 10.15% by introducing nanopatterned Si light-trapping substrates [10]. Malte et al. reported that nanocrystalline SiC could be used to design a transparent passivating front contact in Si-based solar cells, which finally enabled good passivation and high conductivity translating into an improved short-circuit current density (40.87 mA cm$^{-2}$) and fill factor (80.9%) with an efficiency of 26% [11]. Furthermore, the Si/C ratio in SiC films has a significant influence on various properties, such as chemical bonding, the refractive

index, and the absorption coefficient. To control the photoelectric characteristics, Liu et al. adjusted the Si/C ratio and B doping concentration to manipulate the size and distribution of Si quantum dots in SiC films [12]. Moreover, optimizing the Si/C ratio can enhance the efficiency of solar energy absorption and photoelectric conversion in Si-based solar cells. To improve the performance of SiC-based devices, it is crucial to investigate their fundamental electronic properties, particularly their carrier transport properties. However, the conductivity of a SiC matrix is typically low, around $1.9 \times 10^{-10}$ S·cm$^{-1}$ for *a*-SiC film, due to its wide bandgap. This limitation significantly impedes carrier transport [13,14]. As we know, the carrier transport process is complex due to various factors, including grain size, grain boundaries, crystallinity, and interface state, that can impact carrier transport behaviors [15–20]. Therefore, it is imperative to conduct a comprehensive study on the carrier transport properties of SiC films, but this remains a significant challenge.

In the previous work, highly conductive films consisting of P-doped Si NCs embedded in a SiC host matrix (Si NC : SiC films) could be achieved by annealing P-doped amorphous Si-rich SiC films at 1000 °C [21]. According to that report, increasing the doping ratio led to a gradual increase in conductivity, indicating the activity of dopants in the formation of Si NCs. In samples with a low doping ratio, mobility increased with temperature, suggesting the influence of grain boundaries and ionized impurities. In contrast, heavily doped films with a high doping ratio exhibited metal-like behavior, where mobility decreased with temperature due to the dominance of the photon-scattering mechanism in the carrier process. In this study, we have prepared annealed SiC films with varying Si/C ratios at different temperatures and examined their electronic properties, with a focus on Hall mobility. Our findings have indicated that increasing the annealing temperature or Si/C ratio improved the crystallinity of the films, resulting in an increase in room-temperature Hall mobility. Additionally, we have investigated the temperature dependence of conductivity, and we briefly discuss the role of grain boundary scattering mechanisms in conjunction with the height of the potential barrier at the grain boundary.

## 2. Experiment

Hydrogenated amorphous SiC films with various Si/C ratios were prepared with a plasma-enhanced chemical vapor deposition (PECVD) system using gaseous mixtures of pure silane (SiH$_4$, Nanjing, China), methane (CH$_4$, Nanjing, China), and hydrogen (H$_2$, Nanjing, China). The flow rate of SiH$_4$ was kept at 5 sccm (standard cubic centimeters per minute) during the growth process. The Si/C ratio, $R_{Si/C}$, which can be defined as the gas ratio of [SiH$_4$] to [CH$_4$], was changed by adjusting the flow rate of CH$_4$, which was controlled at 1 sccm, 2.5 sccm, and 5 sccm, respectively. During the deposition, the gas-chamber pressure, substrate temperature, and radio-frequency power were kept at 10 mTorr, 250 °C, and 30 W, respectively. All the samples were subsequently annealed in an annealing furnace at different chosen temperatures, namely, 800 °C, 900 °C, and 1000 °C, for 1 h for crystallization after deposition. The substrates chosen for various measurements were quartz plates and monocrystalline Si wafers.

We obtained the Raman spectra of all samples using a Raman spectrometer (model: HR800, Jobin Yvon Horiba Inc., Paris, France). Fourier-transform infrared (FTIR) spectroscopy (model: Nexus 870, Nicolet Inc., Markham, ON, Canada) was used to assess the structural changes and bonding configurations of the samples. The optical bandgaps of the samples were determined using Tauc plots based on the optical absorption spectra measured with a UV spectrophotometer (model: UV-3600, Shimadzu Inc., Kyoto, Japan). The electronic properties of SiC films were studied using a variable-temperature Hall effect measurement system (model: LakeShore 8400 series, LakeShore, Westerville, OH, USA) over a temperature range of 300 K to 480 K with a 20 K step size. All samples were prepared with coplanar Al electrodes on the four corners using vacuum thermal evaporation, followed by a 30-min alloying treatment at 400 °C to achieve ohmic contacts.

## 3. Results and Discussion

### 3.1. Microstructure Characterization

The microstructures of SiC films with various Si/C ratios annealed at different temperatures were initially studied by Raman spectroscopy. The Raman spectra of SiC films with $R_{Si/C} = 1$ and 5 before and after annealing at 800 °C, 900 °C, and 1000 °C were compared as shown in Figure 1a,b. A broad peak was found near the 480 cm$^{-1}$ wave number in the as-deposited samples, indicating an amorphous Si-Si phase in the microstructures of the amorphous SiC films. After thermal annealing, a sharp Raman peak around the 520 cm$^{-1}$ wave number, demonstrating a crystalline Si structure, appeared in the annealed samples and became more and more intense with increasing annealing temperatures. Meanwhile, the intensity of the Raman peak around the 520 cm$^{-1}$ wave number was also gradually enhanced via an increase in $R_{Si/C}$, as shown in Figure 1c. Our findings suggest that Si nanocrystals can be crystallized in amorphous SiC films during the annealing process. As we all know, the crystalline volume fraction ($X_c$) of annealed samples can usually be deduced by the formula $X_c = I_c/(I_c + I_a)$, where $I_a$ and $I_c$ are the amorphous and crystalline parts of the integrated Raman scattering intensity, respectively [22]. It can be easily inferred that the crystallization of Si nanocrystals in SiC films can be improved by increasing the annealing temperature and the Si/C ratio as well.

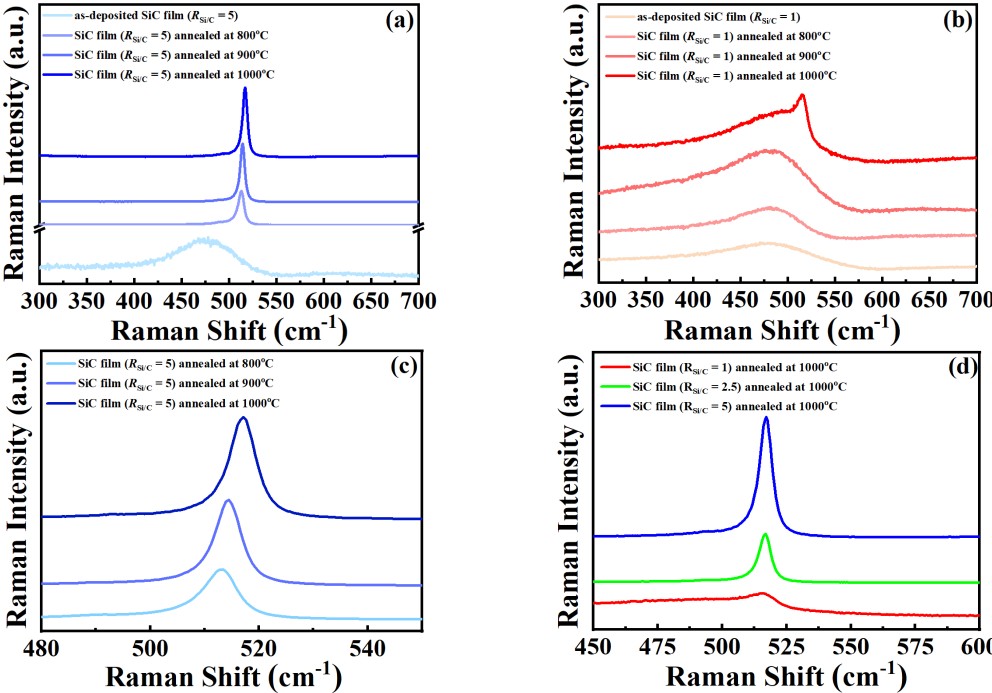

**Figure 1.** The Raman spectra of SiC films with $R_{Si/C} = 1$ (**a**) and 5 (**b**) before and after annealing at 800 °C, 900 °C, and 1000 °C. (**c**) The Raman spectra of SiC films with $R_{Si/C} = 5$ annealing at 800 °C, 900 °C, and 1000 °C and (**d**) The Raman spectra of SiC films with various $R_{Si/C}$ values annealed at 1000 °C.

Furthermore, the position of the Raman peaks around 520 cm$^{-1}$ in all annealed samples had slightly downshifted with respect to the peak position of monocrystalline Si (520 cm$^{-1}$). The mean size of Si nanocrystals in the annealed samples could be calculated according to the empirical formula $D_R = 2\pi\sqrt{B/\Delta\omega}$, where $D_R$ is the average grain size of Si nanocrystals, $\Delta\omega$ is the Raman shift of the crystalline peak from the monocrystalline Si peak, and $B$ is set at 2.24 cm$^{-1}$ for Si on the basis of previous reports [23]. It was found that the value of $\Delta\omega$ was decreased from 6.1 cm$^{-1}$ to 2.6 cm$^{-1}$ in the SiC films with $R_{Si/C} = 5$ when the annealing temperature increased from 800 °C to 1000 °C, as shown in Figure 1c. Accordingly, $D_R$ was calculated to be increased from 4 nm to about 6 nm as the

annealing temperature increased. However, it should be pointed out that the positions of Raman peaks around 520 cm$^{-1}$ were consistent in the annealed samples with various $R_{Si/C}$ values annealed at a certain temperature, as shown in Figure 1d, which implies that the grain size of Si nanocrystals was not greatly changed with Si/C ratios. Based on the Raman results, it can be inferred that thermal annealing aids in the growth of Si nanocrystals. The annealing temperature appears to have a significant impact on the grain size of the Si nanocrystals in SiC films, with an increase in annealing temperature resulting in a gradual increase in grain size. On the other hand, as the Si/C ratio increases, the number of Si nanocrystals generated in the SiC films also increases. During the annealing process, these nanocrystals are discretely spread over the films, contributing to the crystallization of Si nanocrystals in the films. Both of these factors play crucial roles in the crystallization of SiC films.

The FTIR spectra of SiC films with various Si/C ratios before and after annealing at 1000 °C are shown in Figure 2. We found that the absorption bands at 640 cm$^{-1}$ and 2000 cm$^{-1}$, corresponding to the wagging mode of the silicon hydride SiH$_n$ and the stretching mode of H-Si-Si$_3$ [24], could be clearly identified in the as-deposited samples. The above two absorption bands measured in the films suggest that almost all of the hydrogen had been bonded to Si in the form of Si–H. However, both of the abovementioned absorption bands were found to have vanished after annealing at 1000 °C, which implies that the hydrogens were gradually effused out of the films during the annealing process. Finally, nanocrystalline Si grains were formed increasingly after the annealing temperature exceeded 800 °C, and this conclusion was consistent with the Raman results.

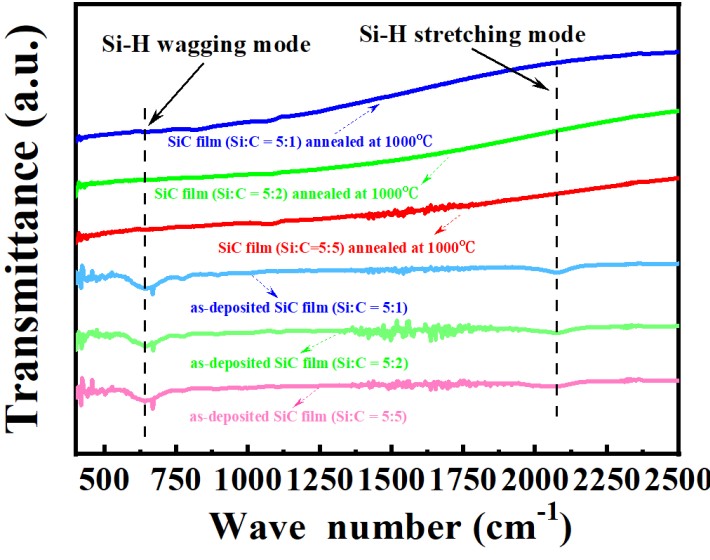

**Figure 2.** The FTIR spectra for SiC films with various Si/C ratios before and after annealing at 1000 °C.

Measurements were conducted for the reflection and transmission of SiC films before and after thermal annealing in order to explore the optical performance of SiC films. The optical bandgap, $E_g$, of SiC films with various Si/C ratios before and after annealing at 1000 °C can be deduced by using the Tauc plot as shown in Figure 3. The plots of $(\alpha h\nu)^{1/2}$ versus photon energy $h\nu$ in Figure 3a show that the optical bandgaps of as-deposited samples with various Si/C ratios were around 1.75 eV, representing the bandgaps of amorphous SiC films. As shown in Figure 3b, we noted that the optical bandgaps of SiC films after 1000 °C annealing were obviously increased, reaching about 2.10 eV with $R_{Si/C} = 1$. It is generally believed that the structures of SiC films are changed from amorphous to nanocrystalline phases via thermal annealing. During crystallization, disordered grain boundary regions that have a higher optical gap than the amorphous SiC regions are gradually generated, thus contributing to the overall optical gap of the SiC films after

annealing [25]. Therefore, the increase in the optical gap for the SiC films annealed at 1000 °C can be mainly ascribed to the increase in nanocrystalline components in the films. This can also explain the increase in the optical bandgaps with $R_{Si/C}$, which reached 2.23 eV in the SiC film with $R_{Si/C} = 5$. Furthermore, the increase in the optical bandgap may also be caused by the incorporation of N or O into the annealed films during the annealing process, as usually observed in previous studies [26].

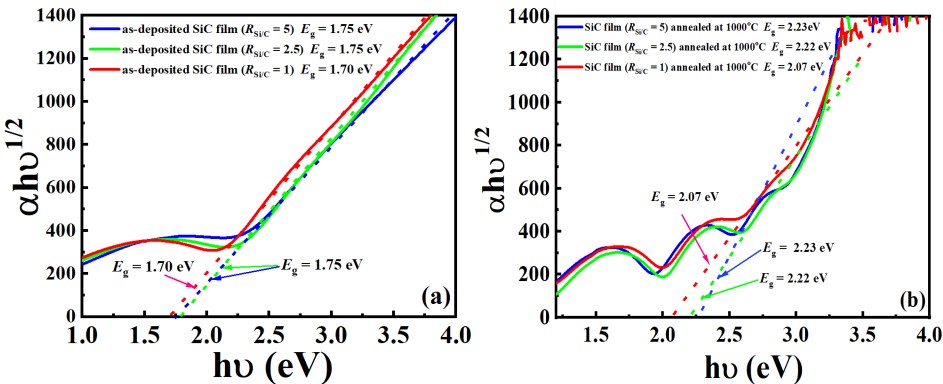

**Figure 3.** The plots of $(\alpha h\upsilon)^{1/2}$ versus photon energy $h\upsilon$ for (**a**) the as-deposited SiC films with various $R_{Si/C}$ values and (**b**) the SiC films with various $R_{Si/C}$ values annealing at 1000 °C.

*3.2. Room-Temperature Dark Conductivity*

In order to comprehend the electronic properties of SiC films with various Si/C ratios annealed at different temperatures, we conducted measurements of room-temperature dark conductivity using the Hall effect. As presented in Table 1, it is evident that dark conductivities increased with both annealing temperature and $R_{Si/C}$, reaching $5.4 \times 10^{-7}$ S cm$^{-1}$ in the SiC film with $R_{Si/C} = 5$ annealed at 1000 °C. This conductivity value aligns with our previous investigation of Si NC films. It is known that dark conductivity values are determined by both mobility and carrier concentration. In this study, although there was an enhancement in mobility (further explored in the following sections), its impact on conductivity was relatively modest. Therefore, the improved dark conductivities with increased annealing temperatures and $R_{Si/C} = 5$ can be primarily attributed to the increased carrier density resulting from the formation of nanocrystalline Si phases [18]. Annealing at high temperatures and increasing $R_{Si/C}$ further promoted crystallization, causing the conductivity to increase by at least three orders of magnitude compared to that of the SiC films with lower $R_{Si/C}$ values annealed at relatively low temperatures.

**Table 1.** The values of room temperature-dark conductivity for the SiC films with various Si/C ratios annealed at different temperatures.

| SiC Film | The Room-Temperature Dark Conductivity | | |
| --- | --- | --- | --- |
| | **Annealed at 800 °C** | **Annealed at 900 °C** | **Annealed at 1000 °C** |
| $R_{Si/C} = 1$ | $9.0 \times 10^{-10}$ S cm$^{-1}$ | $2.0 \times 10^{-9}$ S cm$^{-1}$ | $1.4 \times 10^{-8}$ S cm$^{-1}$ |
| $R_{Si/C} = 2.5$ | $3.2 \times 10^{-9}$ S cm$^{-1}$ | $8.9 \times 10^{-9}$ S cm$^{-1}$ | $9.2 \times 10^{-8}$ S cm$^{-1}$ |
| $R_{Si/C} = 5$ | $8.7 \times 10^{-8}$ S cm$^{-1}$ | $1.4 \times 10^{-7}$ S cm$^{-1}$ | $5.4 \times 10^{-7}$ S cm$^{-1}$ |

Another potential factor influencing conductivity is the presence of background dopants. N and O typically serve as background dopants during the annealing process, contributing to increased dark conductivities in microcrystalline silicon films [19,20]. Previous reports indicated that background doping with N or O led to an increase in free carriers, with estimated doping levels on the order of $10^{17}$~$10^{20}$ cm$^{-3}$. However, in our case, the carrier concentration in the SiC films was approximately $10^{11}$~$10^{12}$ for the

annealed samples, as estimated from the numerical values of conductivity and mobility. This concentration is much lower than that reported in the current research. Therefore, we posit that the incorporation of N or O into the annealed films during the annealing process did not significantly impact the conductivity of the SiC films.

### 3.3. Room-Temperature Hall Mobility

As is well known, the measurement of Hall mobility, $\mu_{\text{Hall}}$, is commonly used to study the electronic properties of semiconductor materials, particularly the carrier transport mechanisms. Figure 4 shows the Hall mobility measured at room temperature in the films with various $R_{\text{Si/C}}$ values annealed at different temperatures. The data showed that the Hall mobility in the SiC film with $R_{\text{Si/C}} = 1$ annealed at 800 °C was only about 0.2 cm$^2$/V·s. According to the characterization of the microstructure, we confirmed that the crystallinity of this film ($R_{\text{Si/C}} = 1$, 800 °C annealing) was relatively low; therefore, the value of 0.2 cm$^2$/V·s refers to the Hall mobility in an amorphous SiC film at room temperature. During the annealing process, the Hall mobilities gradually increased with the annealing temperature as well as $R_{\text{Si/C}}$. In Figure 4, it is evident that the $\mu_{\text{Hall}}$ value exhibited an increase from 0.2 cm$^2$/V·s (achieved with $R_{\text{Si/C}} = 1$ and annealing at 800 °C) to 0.8 cm$^2$/V·s with a rise in the annealing temperature to 1000 °C. Furthermore, the $\mu_{\text{Hall}}$ value further escalated to 2.1 cm$^2$/V·s when the $R_{\text{Si/C}} = 5$. These increased Hall mobilities can be attributed to the continual formation of Si NCs in the amorphous SiC films. Unlike carrier transport behavior in amorphous SiC films, transport behaviors in annealed SiC films are mainly determined by carrier transport in the inner of Si NCs, which usually presents a high Hall mobility compared with that in amorphous SiC films, and hopping conduction from one Si nanocrystal to another one [26]. Thus, annealed SiC films with high crystallinity usually exhibit higher Hall mobility. Similarly, we also observed an increase in mobility with $R_{\text{Si/C}}$ in the annealed SiC films (as shown in Figure 4), which can still be attributed to the improvement of crystallinity in the films.

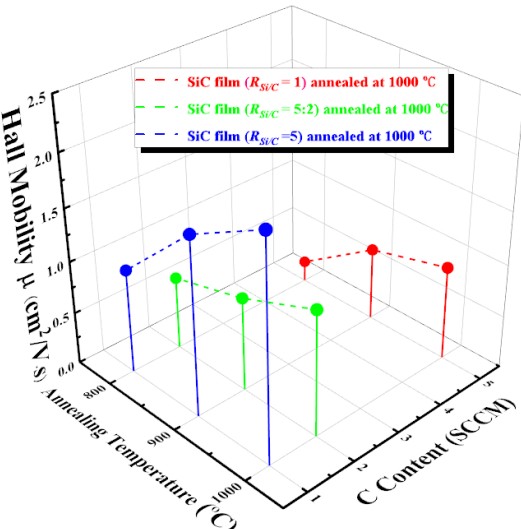

**Figure 4.** Room-temperature Hall mobility for the SiC films with various Si/C ratios annealed at different temperatures.

### 3.4. Temperature-Dependent Hall Mobility

In order to deeply investigate the electronic properties of SiC films, especially the carrier transport behaviors, temperature-dependent Hall measurements in the SiC films with various $R_{\text{Si/C}}$ values annealed at different temperatures were performed as shown in Figure 5a–c. It should be pointed out that Hall mobilities ($\mu_{\text{Hall}}$) as a function of test temperature ($T$) show different behaviors in different films. As shown in Figure 5a, the Hall mobility in the SiC film with $R_{\text{Si/C}} = 1$ annealed at 800 °C, which showed an extremely low value of 0.2 cm$^2$/V·s at room temperature, was slightly decreased with increasing

test temperature. As mentioned above, we can approximate the SiC film with $R_{Si/C} = 1$ annealed at 800 °C as an *a*-SiC film. In amorphous thin films, the carrier Hall mobility is greatly limited due to the large number of defects, the presence of impurities, and the amorphous structure, which result in relatively low Hall mobility [27–29]. The Hall mobility of *a*-SiC films tends to decrease as temperature increases due to the greater scattering of charge carriers and the increased number of scattering centers within the film. This leads to a reduction in Hall mobility.

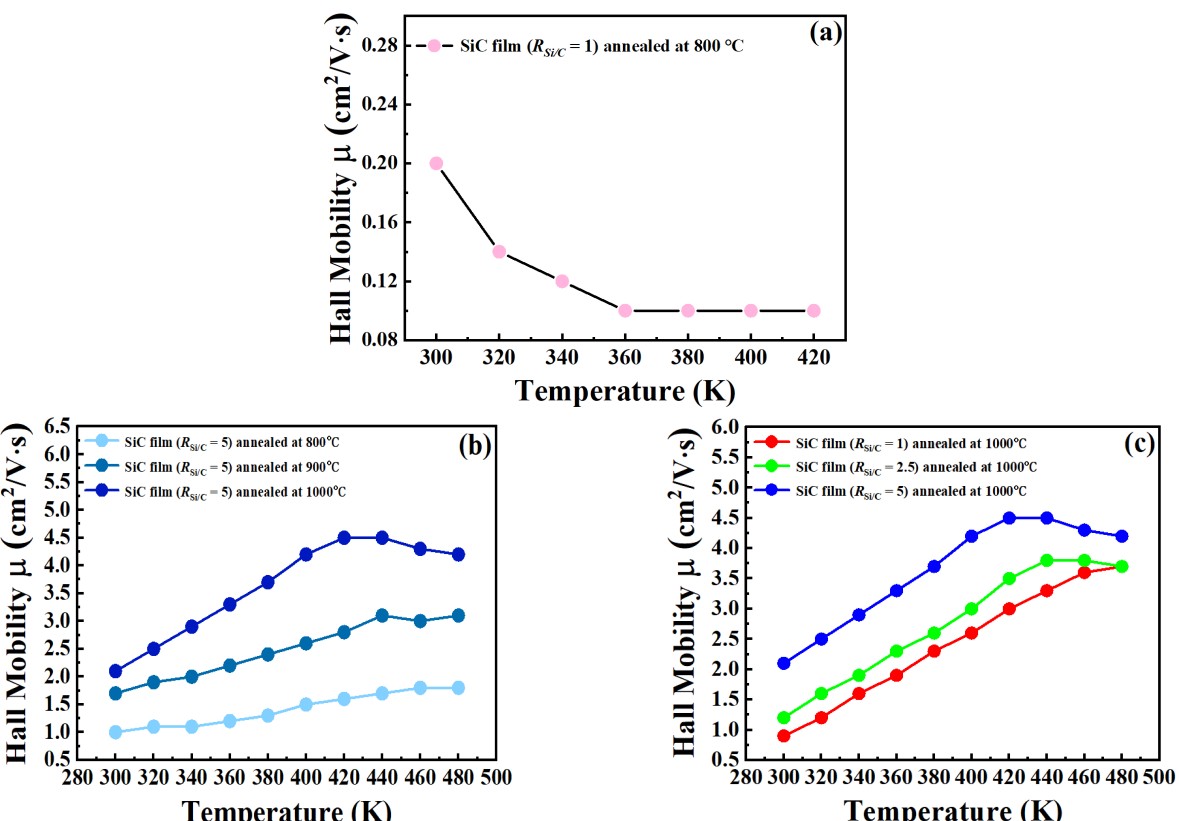

**Figure 5.** The temperature-dependent Hall mobility of (**a**) the SiC film with $R_{Si/C} = 1$ annealed at 800 °C; (**b**) the SiC films with $R_{Si/C} = 5$ annealed at 800 °C, 900 °C, and 1000 °C; and (**c**) the SiC films with various $R_{Si/C}$ values annealed at 1000 °C.

Figure 5b shows the $\mu_{Hall}{\sim}T$ curves in the SiC films with $R_{Si/C} = 5$ annealed at 800 °C, 900 °C, and 1000 °C. Meanwhile, Figure 5c shows them for the SiC films annealed at 1000 °C with $R_{Si/C} = 1$, 2.5, and 5. It was found that the Hall mobilities of all the annealed films increased with increasing measured temperature, which is exactly opposite to the behavior of *a*-SiC films. In our previous reports, this behavior was often used to explain the grain-boundary (GB) scattering that occurred during the transport of charge carriers [25]. Under the GB scattering mechanism, the charge carriers need to acquire enough energy to overcome the potential barrier at the GBs during the transport process. The higher the measured temperature, the more energy the carriers obtain, and the easier it is for them to cross over the potential barrier. As a result, the behavior in which the carrier mobility increases with measured temperature is observed.

Seto's model is commonly used to describe the GB scattering process, and the mobility shows thermally activated behavior as follows:

$$\mu_{Hall} = \mu_0 exp\left(-E_B^\mu / kT\right) \tag{1}$$

where $\mu_0$ is the exponential prefactor and $E_B^\mu$ is the activation energy that corresponds to the potential energy barrier height at the GBs [30]. As shown in Figure 6, a good linear

relationship between $ln\mu_{Hall}$ and $1000/T$ indicated that the experimental results were well fitted with Equation (1). The value of $E_B^\mu$ can be deduced from the slope of the line of fit, as shown in Table 1. For the SiC films with $R_{Si/C} = 5$ annealed at different temperatures, we can clearly identify that the barrier height $E_B^\mu$, which was about 21 mV in the film annealed at 800 °C, increased gradually as the annealing temperature increased. The value of $E_B^\mu$ finally reached 34 mV for the film annealed at 1000 °C. From the relationship between the $E_B^\mu$ and the annealing temperature $T$, we can easily understand that the increase in the potential energy barrier height at GBs may be related to the improvement of crystallization in films. As the annealing temperature increases, the grains in the films gradually grow, and the size and number of grains gradually increase, forming a large number of GBs. Thus, the appearance of GBs is then followed by an increase in the barrier height at GBs in the annealed SiC films.

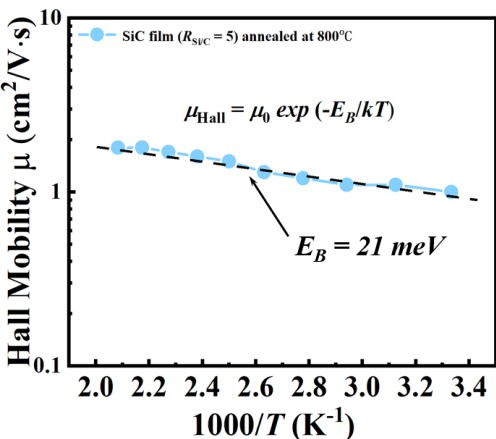

**Figure 6.** The relationship between $ln\mu_{Hall}$ and $1000/T$ for the SiC film with $R_{Si/C} = 5$ annealed at 800 °C.

Interestingly, however, we found that there was almost no change in the value of $E_B^\mu$ when analyzing the annealed SiC films with various $R_{Si/C}$ values as shown in Table 2. From the previous discussion, we have established that improving $R_{Si/C}$ can also improve crystallization in the annealed SiC films. It is very strange that the improvement in crystallization in this case did not cause a significant rise in barrier height in the films. Although increasing the annealing temperature or the Si/C ratio can improve crystallization in annealed SiC films, we tried to find some differences from them according to the Raman results. In the former, the improvement of crystallinity in annealed SiC films was mainly manifested in the increase of the number of nanocrystals as well as the growth of nanocrystals grain size. As for the latter, we found that the grain size of nanocrystals did not increase significantly, and the improvement in the crystallization was mainly contributed to by the increase in the number of nanocrystals in annealed SiC films. Seto et al. proposed a model when studying the electrical properties of microcrystalline silicon [30]. In the Seto model, the grain size of microcrystalline silicon is denoted as $L$ cm, and the carrier concentration in the material is $N$ cm$^{-3}$. The thickness of GBs is negligible compared to the grain size, and there is a "trap"-state surface density of $N_t$ cm$^{-2}$ at the GBs. Due to the capture of charge carriers, a depletion region is formed in the GBs. Figure 7 illustrates a band diagram of the Seto model.

**Table 2.** The values of $E_B^\mu$ for the SiC films with various Si/C ratios annealed at different temperatures.

| The Potential Energy Barrier Height at GBs | | | |
|---|---|---|---|
| SiC Film | Annealed at 800 °C | Annealed at 900 °C | Annealed at 1000 °C |
| $R_{Si/C} = 1$ | - | - | 32 meV |
| $R_{Si/C} = 2.5$ | - | - | 32 meV |
| $R_{Si/C} = 5$ | 21 meV | 26 meV | 34 meV |

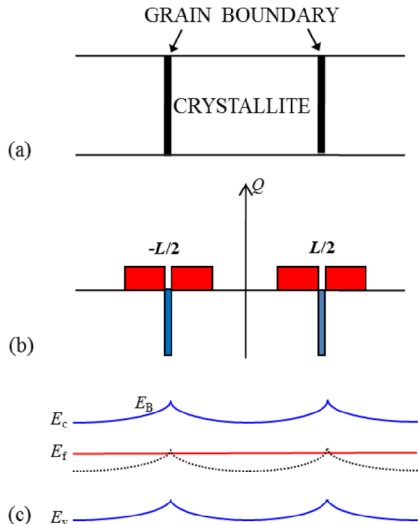

**Figure 7.** The band diagram of the Seto model (**a**) model for the crystal structure; (**b**) The charge distribution within the crystallite and at the grain boundary; (**c**) the energy band. Adapted from [30].

Under this assumption, two situations should be considered. One is that the defect-state concentration at the GBs is higher than the carrier concentration in the material. The other is that the carrier concentration in the material is higher than the defect-state concentration at the GBs. In this way, the relationship between barrier height and carrier concentration can be expressed by the following equation:

$$E_B = \begin{cases} \frac{e^2 L^2 N}{8\varepsilon} & N_t > NL \\ \frac{e^2 N_t^2}{8\varepsilon N} & N_t < NL \end{cases}$$

where $e$ is the charge of an electron and $\varepsilon$ is the dielectric constant of Si. Considering the grain size $L$ of about 10 nm, the defect surface concentration $N_t$ at the GBs of Si NCs is on the order of $10^{12}$ cm$^{-2}$, and the carrier concentration in the current work is on the order of about $10^{12}$ cm$^{-3}$, which satisfies the condition $N_t > NL$. Accordingly, $E_B = \frac{e^2 L^2 N}{8\varepsilon}$ is chosen for this situation. We can easily find that the barrier height of GBs ($E_B$) increases with the grain size of Si NCs. From the discussion above, we conjecture that the average height of the potential barrier at GBs in the annealed SiC films is mainly determined by the grain size of nanocrystals rather than the grain number.

## 4. Conclusions

In summary, we fabricated hydrogenated amorphous SiC films with varying Si/C ratios using PECVD methods and then subjected them to thermal annealing at different temperatures to induce crystallization. Microstructure characterization revealed that increasing the annealing temperature promoted an increase in individual grain size, while increasing the Si/C ratio resulted in a higher number of grains. Both of them contributed to crystallization in the annealed SiC films. The electronic properties, especially the room-temperature dark conductivities and Hall mobilities, were investigated, and it was found that both the room-temperature dark conductivities and Hall mobilities in annealed SiC films were increased after an increase in the annealing temperature or Si/C ratio, which can be ascribed to the improvement of crystallinity in the films. The formation of nanocrystalline Si in the SiC film contributed to the increase in the carrier concentration, thereby enhancing the dark conductivity of the film. Simultaneously, the formation of nanocrystalline Si also contributed to the transport of charge carriers, thus improving the mobility in film. Based on temperature-dependent Hall effect measurement, the scattering mechanisms during the carrier transport process in SiC films were investigated. It should be noted

that the grain-boundary scattering behaviors were observed in SiC films with various Si/C ratios annealed at different temperatures. The potential energy barrier height at the grain boundaries was calculated using the $\mu_{Hall} \sim T$ relationship. Notably, increasing the annealing temperature had a greater impact on the potential barrier height than increasing the Si/C ratio in SiC films. This suggests that changes in grain size are more likely to affect the height of the potential barrier at the grain boundaries in annealed SiC films.

**Author Contributions:** D.S. (Dan Shan) and Y.C. conceived the idea and carried out the experiments. M.W. and D.S. (Daoyuan Sun) participated in the preparation of the samples. D.S. (Dan Shan), M.W., and Y.C. took part in the experiments and the discussion of the results. D.S. (Dan Shan) and Y.C. drafted the manuscript. All authors have read and agreed to the published version of the manuscript.

**Funding:** This work was supported by the Major Project of the Natural Science Foundation of Education Department in Jiangsu Province (22KJA510008), the Science and Technology Planning Project of Yangzhou City (YZ2022209 and YZ2021132), the New Perception Technology and Intelligent Scene Application Engineering Research Center of Jiangsu Province (2021), the Carbon-Based Low-Dimensional Semiconductor Materials and Device Engineering Research Center of Jiangsu Province (2023), and the Jiangsu Province Vocational Education Wisdom Scene Application "Double Teacher" Master Teacher Studio (2021).

**Data Availability Statement:** No new data was created or analyzed in this study. Data sharing is not applicable to this article.

**Conflicts of Interest:** The authors declare no conflicts of interest.

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
