# Peer review of "Investigation of the Electronic Properties of Silicon Carbide Films with Varied Si/C Ratios Annealed at Different Temperatures"

_crystals, doi:10.3390/cryst14010045_

Round 1

Reviewer 1 Report

Comments and Suggestions for Authors

The paper with title “The Investigation of Electronic Properties in Silicon Carbide Films with Varied Si/C Ratios Annealed at Different Temperatures” presents an interesting study related to the electronic properties of SiC.

There are some minor issues that should be addressed:

  1. In the “Introduction” part, the authors should underline the novelty of their work because there are numerous studies related to their topic.
  2. Also, reference [21] is about SnTe not about a previous work. Please correct this issue and check all the references in the text.
  3. In the “Results and Discussion” part, the figure 1a,b should be grouped with 1c,d. Otherwise, fig. 1c,d should be renumbered, becoming fig 2a,b, and also all the figures in the manuscript will be renumbered.
  4. What is the thickness of the SiC films? How the Hall mobility is influenced by the thickness of SiC films?
  5. The authors are talking about the microstructure of SiC films but there is no SEM or AFM image of the studied films. Can you add some data and discuss about that?

 My decision: minor revision.

Comments on the Quality of English Language

The English language can be improved.

Reviewer 2 Report

Comments and Suggestions for Authors

The Article is devoted to the study of films of hydrogenated amorphous SiC (a-SiC:H) with different Si/C ratios. The results of measurements using Raman spectroscopy, infrared and UV-VIS absorption of SiC of films annealed at different temperatures, as well as their electronic properties obtained from temperature-dependent Hall mobility are presented. Different temperature behavior of the Hall mobility in films was discovered depending on the Si/C ratio and annealing temperature. The transfer processes in SiC films are considered taking into account the grain boundary scattering.

Since SiC films are currently of great practical importance in various fields and attract the attention of researchers, obtaining new information about their properties is undoubtedly of interest. The novelty of the research carried out is manifested in the study of the electronic properties of Si/C films depending on the Si/C ratio and the discovery of the dependence of the Hall mobility on the parameters of the film and its processing. The list of references sufficiently reflects the current state of the art. The style of presentation and English are good enough.

Notes:

1. Seems to require a little more information about the substrates (roughness, thickness, dimensions, etc.).

2. It seems that the increments on the axes Fig.5 are too small.

Conclusion: The Article is of interest to the journal and can be published as it is.

Reviewer 3 Report

Comments and Suggestions for Authors

The Investigation of Electronic Properties in Silicon Carbide Films with Varied Si/C Ratios Annealed at Different Temperatures

a-SiC:H films were deposited on different substrates by using a PECVD technique at different Si/C ratios. Afterward, the samples were annealed at different temperatures to induce crystallization.

The authors demonstrated that the Hall mobilities of the SiC films increase with the annealing temperature and its Si/C ratio which is attributed to the crystallization processes of the SiC films.

The authors described the transport mechanism in the films with a particular emphasis on the grain boundary scattering.

The article is interesting and well-written, however, there is some observation and sites to be corrected.

1.       Please rearrange the references to be the corresponding number from the text.

2.       Let know for what purpose the references 33 through 38?

3.       In the paragraph 120-126, the authors intuit the size of the nanocrystal.

How close or real are the calculated nanocrystalline sizes? Is there any evidence, for example, by using an HRTEM image for the grown SiC nanocrystalline sizes?

4.       Please explain the possible formation of SiC-nanocrystalline, instead of only Si-NCs embedded in an a-SiC.

5.       In rows 166-167, the authors mentioned that: “the increase in the optical gap for the SiC films annealed at 1000℃ is mainly ascribed to the increase of nanocrystalline components”

What is the expected nanocrystalline size (the increase of its component) after such a high temperature annealing treatment ?

6.       In rows 183 to 185, the authors mentioned that: “During the annealing process, the Hall mobilities were gradually increased with annealing temperature, which can be attributed to the continual formation of Si NCs in the amorphous SiC films”.

It is hard to understand how the Si NCs will continuously be produced due to the annealing processes. Is it regards to the amount of Si NCs itself, or related to its size?

7.       In rows 237 to 240 and from 260 and 261, the authors mentioned that: “by the grain size of nanocrystals rather than the grain number. The larger the nanocrystals, the more distinct the boundaries between them, and the higher the potential barrier at the grain boundaries”

Please explain, how the potential barrier relates to the distinct boundaries between the nanocrystals. It is hard to understand if increases its potential barrier with the grain's growth. i.e. the GB growth supposes less amount of nanocrystalline, and then, the GBs-related area should be reduced.

Comments on the Quality of English Language

I think the english language is good enough

Round 2

Reviewer 3 Report

Comments and Suggestions for Authors

Please, check the references 27 to 30 if is the correct and if is indicated in the text.

Author Response

Thank you for your correction. I have made the modifications in the original text.